# Role of Texture Analysis in Oropharyngeal Carcinoma: A Systematic Review of the Literature

**DOI:** 10.3390/cancers14102445

**Published:** 2022-05-16

**Authors:** Eleonora Bicci, Cosimo Nardi, Leonardo Calamandrei, Michele Pietragalla, Edoardo Cavigli, Francesco Mungai, Luigi Bonasera, Vittorio Miele

**Affiliations:** 1Department of Experimental and Clinical Biomedical Sciences, Radiodiagnostic Unit n. 2, University of Florence—Azienda Ospedaliero-Universitaria Careggi, Largo Brambilla 3, 50134 Florence, Italy; cosimo.nardi@unifi.it (C.N.); leonardo.calamandrei@unifi.it (L.C.); michelepietragalla2@gmail.com (M.P.); 2Department of Radiology, University of Florence—Azienda Ospedaliero-Universitaria Careggi, Largo Brambilla 3, 50134 Florence, Italy; edoardocavigli@yahoo.it (E.C.); f.mungai@gmail.com (F.M.); luigi.bonasera72@gmail.com (L.B.); vmiele@sirm.org (V.M.)

**Keywords:** radiomics, texture analysis, tonsil, oropharynx, head and neck, squamous cell carcinoma

## Abstract

**Simple Summary:**

The incidence of squamous cell carcinomas of the oropharynx has rapidly increased in the last two decades due to human papilloma virus infection (HPV). HPV-positive and HPV-negative squamous cell tumours differ in radiological imaging, treatment, and prognosis; therefore, differential diagnosis is mandatory. Radiomics with texture analysis is an innovative technique that has been used increasingly in recent years to characterise the tissue heterogeneity of certain structures such as neoplasms or organs by measuring the spatial distribution of pixel values on radiological imaging. This review delineates the application of texture analysis in oropharyngeal tumours and explores how radiomics may potentially improve clinical decision-making.

**Abstract:**

Human papilloma virus infection (HPV) is associated with the development of lingual and palatine tonsil carcinomas. Diagnosing, differentiating HPV-positive from HPV-negative cancers, and assessing the presence of lymph node metastases or recurrences by the visual interpretation of images is not easy. Texture analysis can provide structural information not perceptible to human eyes. A systematic literature search was performed on 16 February 2022 for studies with a focus on texture analysis in oropharyngeal cancers. We conducted the research on PubMed, Scopus, and Web of Science platforms. Studies were screened for inclusion according to the preferred reporting items for systematic reviews. Twenty-six studies were included in our review. Nineteen articles related specifically to the oropharynx and seven articles analysed the head and neck area with sections dedicated to the oropharynx. Six, thirteen, and seven articles used MRI, CT, and PET, respectively, as the imaging techniques by which texture analysis was performed. Regarding oropharyngeal tumours, this review delineates the applications of texture analysis in (1) the diagnosis, prognosis, and assessment of disease recurrence or persistence after therapy, (2) early differentiation of HPV-positive versus HPV-negative cancers, (3) the detection of cancers not visualised by imaging alone, and (4) the assessment of lymph node metastases from unknown primary carcinomas.

## 1. Introduction

Head and neck cancers represent around 3% of all malignancies and squamous cell carcinoma is the most frequent histotype (85%) [1,2].

The incidence of squamous cell carcinomas of the oropharynx (OPSCC) has been rapidly increasing in the last two decades due to human papilloma virus (HPV) infection that is associated with the development of carcinomas that primarily involve lingual and palatine tonsils [3].

HPV-positive (HPV+) squamous cell tumours mainly affect males with a mean age of 40–60 years, and independent of risk factors such as tobacco exposure or alcohol consumption. Some HPV genotypes are considered oncogenic including genotypes 16 and 18. These are the most frequent ones [4].

The diagnosis of OPSCC is confirmed by a pan-endoscopy with multiple biopsies to search for the primary lesion [5]. Histological examination is the gold standard tool for the characterisation of the lesion and the detection of viral DNA or transcription products [6].

HPV-negative (HPV−) tumours occur more frequently in the elderly population (over 70 years of age). The most significant risk factors for these types of cancer are smoking and alcohol consumption [7,8].

HPV+ and HPV− squamous cell tumours differ in radiological imaging, treatment, and prognosis; therefore, differential diagnosis is mandatory [9]. At computed tomography (CT) and magnetic resonance imaging (MRI) HPV+ primary tumours are more likely to show enhancement with well-defined borders and exophytic growth, whereas HPV− tumours often show ill-defined borders [10,11,12].

Cervical cystic and necrotic nodal metastases are associated with HPV+ and HPV− OPSCC, respectively [11]. MRI is the best technique to depict cystic or necrotic lymph node metastases and to evaluate soft tissues, tumour margins, and nerve involvement [13,14,15,16,17].

Some degree of overlap between HPV+ and HPV− cancers can be found, therefore distinguishing one from the other by the visual interpretation of images is not often easy [18,19].

Radiomics with texture analysis is an innovative technique that has been used increasingly in recent years to characterise the tissue heterogeneity of certain structures such as neoplasms or organs through the extraction of features obtained from the analysis of a region of interest (ROI) on CT, MRI, or positron emission tomography (PET-CT) images by measuring the spatial distribution of pixel values [20,21,22,23,24,25,26,27].

Texture analysis can provide more precise structural information not perceptible to the human eye and not affected by interindividual variability [28,29,30,31,32,33].

Texture analysis is based on the extraction of first, second and higher-order parameters. The first order uses the histogram to study the gray-level distribution frequency within the ROI in order to evaluate the single pixel and not its interactions with adjacent pixels. The second order evaluates how often the intensity of one pixel has a specific relationship to that of another pixel through gray-level co-occurrence matrix (GLCM) measurements. A further way to derive second-order parameters is the gray-level run length matrix (GLRLM) that analyses consecutive pixels with the same intensity in a defined direction.

The higher orders assess differences between pixels and voxels in the context of the entire ROI using a neighborhood gray-tone-difference matrix (NGTDM) by identifying variations within the examined space in gray-level intensity [34,35,36,37].

The application of texture analysis in oropharyngeal tumours may be helpful in the diagnosis, prognosis, and assessment of disease recurrence or persistence after therapy. Its use could lead to the early differentiation of HPV+ versus HPV− cancers, as well as the detection of tonsillar cancer not easily visualised by imaging alone and in patients with lymph node neck metastases from unknown primary squamous cell carcinomas.

## 2. Materials and Methods

### 2.1. Pico Question

The systematic review was based on the following PICO question [38]:

Does the implementation of texture analysis (I) yield more accurate results in the diagnosis and prognostic evaluation (O) of OPSCC (P) than morphological and functional imaging alone (C)?

Literature searches were planned by identifying keywords based on the PICO question above-mentioned.

### 2.2. Literature Searches

We performed literature research in accordance with preferred reporting items for a systematic reviews (PRISMA) statement (registration number 321983) for studies with a focus on texture analysis in oropharyngeal cancers. We conducted the research based on articles written in English-language on PubMed, Scopus, and Web of Science platforms [39]. The following combined terms were investigated: texture analysis, oropharynx, radiomics, squamous cell carcinoma, and head and neck (Table 1). The detailed search strategy is presented in Appendix A.

### 2.3. Inclusion and Exclusion Criteria

Our search included articles published in international peer-reviewed journals based on texture analysis and radiomics of the oropharynx. We also included studies on the head and neck region that contained sections dedicated to the oropharynx. Original articles, case reports, short communications, and letters to the editor were also included. The exclusion criteria were:-articles that did not deal with oropharyngeal tumours;-articles on the oropharynx not concerning cancer pathology;-articles on oropharyngeal carcinomas that did not mention texture analysis and/or radiomics.

### 2.4. Study Selection and Data Extraction

Two reviewers (CN and MP) independently evaluated the titles and abstracts of the articles selected to assess their eligibility. When the abstract was not sufficient to evaluate the content of the article, the entire text was reviewed, as well as in cases where the article was considered eligible for inclusion. Two authors (EB and LC) independently assessed the risk of bias for all included studies in seven different domains: random sequence generation (selection bias), allocation concealment (selection bias), the blinding of participants and personnel (performance bias), the blinding of outcome assessment (detection bias), incomplete outcome data (attrition bias), selective reporting (reporting bias), and other forms of bias, according to the Cochrane method for risk of bias as detailed in the Cochrane Handbook for systematic reviews. Figure A1 and Figure A2 (Appendix B) detail the results [40]. The inter-reader reliability for the selection of the papers and the assessment of risk of bias were calculated using the Cohen kappa value. The extracted data from each study were as follows: (1) anatomic area; (2) number of patients; (3) tumour histotype; (4) imaging technique; (5) type of segmentation—2D or 3D—used for feature extraction; (6) type of software; and (7) features found to be significant. In case there was no agreement among the selected articles, a discussion between the two reviewers was carried out to evaluate their inclusion.

## 3. Results

Twenty-six studies met our eligibility criteria. The steps for selecting and excluding articles are shown in Figure 1. Screening of articles using the different search engines revealed 38 potentially eligible articles for the full-text review. After examining the full text, twelve papers did not meet the inclusion criteria.

The Cohen kappa values showed almost perfect (k = 0.86) and substantial (k = 0.78) agreement between the two readers for the selection of the papers and the assessment of risk of bias, respectively. Overall risk of bias assessment did not highlight specific domains of bias as critical. However, the allocation concealment (selection bias) was the domain at highest risk of bias (7 studies deemed “high risk”) whereas incomplete outcome data (attrition bias) and selective reporting (reporting bias) were the domains of bias at lowest risk (22 studies in both domains were deemed “low risk”) (Figure A1 and Figure A2). 

Among the 26 selected papers (Table 2), the largest number were from the USA [41,42,43,44,45,46,47,48,49,50,51] and Korea [52,53,54,55], whereas the rest of them were from Switzerland [56,57], the Netherlands [58,59], Italy [19,60], Canada [61], Japan [62,63], Taiwan [64,65] and China [45]. Fourteen articles [19,41,43,44,45,46,47,49,52,53,55,57,58,62] had the oropharynx as the anatomic area of interest and seven articles [42,48,54,56,59,60,61] analysed the head and neck district with sections dedicated to the oropharynx. Six, thirteen, and two articles, respectively, used MRI [53,54,55,58,59,61], CT [19,41,42,43,44,45,46,47,52,56,57,60,62], and PET-CT [48,49] as imaging techniques on which texture analysis was performed (Table 3). Among the 26 studies, eleven of them used free software to extract textural data [19,44,50,51,52,53,54,55,58,60,62,63]; five studies used licensed software [43,49,52,61,64], whereas the remaining ten papers used in house developed software not available to the public. Furthermore, ten studies focused on the distinction between HPV+ and HPV- neoplasms [19,41,42,43,44,46,52,57,58,64]. While no specific textural trend could be identified, most studies found histogram features, entropy, and gray-level co-occurrence to be the features more frequently present in radiomic signatures correlated to HPV positivity. Six studies focused on the diagnosis of oropharyngeal neoplasms [49,50,51,54,55,62] developing different complex models that allowed to infer the nature of a lesion with varying levels of sensibility and specificity. However, no specific trend or cutoff emerged from the evaluation of the aforementioned studies. Eleven studies [45,46,47,48,52,56,58,59,60,61,64,65] focused on the development of new prognostic scoring methods that included analysis of textural features. Such studies produced the most complex models, with radiomic signatures composed of up to 2074 different features, consisting of traditional image analysis, clinical features, and radiomics. [51]. Finally, two studies [53,63] focused on the ways to distinguish between squamocellular cancer and lymphoma through differences in textural features. Both studies elaborated models including both texture analysis features and image analysis techniques.

## 4. Discussion

The use of texture features analysis on CT and MRI images in the diagnosis and pre-therapy evaluation of oropharyngeal cancer has become increasingly interesting. Many studies have evaluated whether these innovative techniques can play a role in the differentiation between malignant and benign lesions and in the characterisation of tumour histotypes. Furthermore, given the tremendous prognostic difference between HPV+ and HPV− tumours, great importance could also be obtained in evaluating HPV status and, therefore, in predictive assessment and therapy response.

### 4.1. Use of Texture Analysis in the Evaluation of HPV Status

HPV+ OPSCC generally has a better prognosis than HPV− OPSCC since it is more responsive to radio-chemotherapy. Therefore, these two cancers need to be considered as distinct entities from an epidemiological, histopathological, prognostic, and therapeutic point of view [66,67,68,69].

Recently, in the eighth edition of its *Cancer Staging Manual*, the American joint committee of cancer (AJCC) introduced a new differentiation between HPV+ and HPV− tumour types in terms of both T (primary lesion) and N (lymph nodes) parameters, as well as their stages, in regard to their respective classifications [70].

The presence of p16 and viral DNA is strongly suggestive of HPV+ oropharyngeal cancer [71].

Primary tumours tend to grow submucosally and most often show an expansive and exophytic growth with sharper margins. These features can be found on both CT and MRI images and are related to the histopathological features of the lesion, which are characterised by the growth of basaloid cells with a poor extracellular matrix and no keratinization. On MRI, HPV+ primary lesions have a slightly hyperintense signal in T2-weighted fat-sat sequences and a more homogeneous enhancement after contrast media intravenous injection, compared to HPV− OPSCC. HPV+ lymph node metastases usually appear as a cystic lesion on both CT and MRI images. In addition, HPV+ lymphadenopathies have a homogeneously hypodense central portion and regular margins on CT images [72].

The cystic appearance with homogeneous fluid content is even more evident on MRI, especially on T2-weighted TSE sequences [73].

HPV− oropharyngeal tumours are keratinizing tumours that often present with ulcerative and necrotic changes. Associated lymphadenopathies are solid and show enhancement after intravenous contrast media administration. Areas of intranodal necrosis and wall thickening are also common [74].

Lesion characterisation is still based on histopathological examination, but recent alternative tools such as texture analysis may become helpful not only in the detection of lesions, but also in the characterisation of their HPV status.

Several studies were carried out to discriminate the HPV status of OPSCC by using texture analysis on CT imaging.

In a study by Choi et al. [52], 86 untreated patients were recruited to assess whether a specific texture shape feature named spherical disproportion correlated to HPV positivity on CT imaging. Spherical disproportion is the indicator of shape irregularity since it is the ratio of the ROI surface to the surface of a sphere with the same volume as the ROI. HPV+ tumours were found to show lower values of spherical disproportion due to their physiological greater roundness. Yu et al. [44] selected two features with MeanBreadth, an index of ROI width (closely related to tumour size) and, as in the previous study, SphericalDisproportion. They assessed how HPV+ tumours in relation to their smaller size had a lower MeanBreadth value than HPV−. The study also showed that HPV+ tumours have lower SphericalDisproportion values than HPV− tumours due to their less complex tumour shape.

In the study by Leijenaar et al. [57], selected features were higher in HPV+ than HPV− tumours, especially low-gray-level-large-size-emphasis, representing a lower contrast uptake into the lesion. Instead, other features were lower in HPV+ than HPV−, including GLSZM and small-zone-emphasis, for a possible greater homogeneity of the lesion and GLCM inverse variance, related to more significant intensity variability in adjacent voxels.

The study conducted by Bogowicz et al. [56] on 93 patients with OPSCC also resulted in a good performance in discriminating HPV status in both the training (AUC = 0.85) and the validation cohort (AUC = 0.78) thanks to four features corresponding to standard deviation, small zone high gray-level emphasis, difference entropy, and coefficient of variation.

Ranjbar et al. [43] also assessed histogram features (median and entropy) and GLCM entropy as statistically significant.

Comparable results were obtained by Fujita et al. [42]. They found statistically significant differences in the distinction of HPV+ status in histogram features (mean, median, entropy, skewness, and kurtosis) and in three GLCM features (contrast, correlation, and energy).

Histogram features (entropy and median) and GLCM entropy were also statistically significant in the study by Buch et al. [41] in differentiating HPV status.

In a study by Mungai et al. [19], several higher order parameters were statistically significant. All parameters derived from a neighbouring gray-level tone dependence matrix (NGLDM) analysis showed lower values in HPV+ than HPV− OPSCC, whereas those derived from a gray-level run-length matrix (GLRML) and gray-level zone-length matrix (GLZLM) were higher than in HPV−. In particular, the lower values of NGLDM, representing the intrinsic heterogeneity of the tumour, may be correlated with the micro- and macroscopic aspects of HPV+ primary tumours since they are non-keratinizing lesions with more defined margins and regular growth. In contrast, higher values in HPV+ OPSCC of parameters belonging to the GLRML and GLZLM categories correlated with the high homogeneity.

Haider et al. [50] analysed 435 primary tumours and 741 cervical lymph node metastases using FDG-PET and CT images. They found that the extrapolated data from the individual methods were almost comparable with the highest predictive performance achieved when PET and CT radiomic features were combined (an AUC of 0.78, and 0.77 for the prediction of HPV association using primary tumour lesion features in a cross-validation and independent validation, respectively). Predictive performance was also higher when PET radiomic markers derived both from the primary tumor and metastatic cervical nodes [50].

Regarding MRI, Dang et al. [61] conducted a study to evaluate a predictive model of HPV status based on T1–T2 weighted and DWI sequences on MRI images, evaluating how texture analysis has an accuracy of approximately 80% in predicting the status of OPSCC.

A study by Bos et al. [58] on MRI of 153 patients showed that radiomics is also valuable for the differentiation between HPV+ and HPV− OPSCC due to their different biology. HPV+ primary tumours, being more regular in shape, tend to have radiomic features able to evaluate their rounder appearance as well as lower maximum intensity values and texture homogeneity. In contrast, HPV− primary tumours are more irregular with different grades of intra-lesional differentiation represented by their texture heterogeneity.

### 4.2. Use of Texture Analysis in the Diagnosis of Oropharyngeal Cancer

The study conducted by Kim et al. [49] aimed to look for a correlation between texture parameters in the differentiation between tonsillar cancer and normal tonsillar tissue and to correlate them with ^18^F-FDG PET/CT, in order to investigate the relationship between texture analysis and metabolic parameters. Neoplastic tonsil is characterized by asymmetric enlargement with increased enhancement after contrast media intravenous injection. Conversely, during the early stages of the disease, the radiological appearance of the lesion may be similar to the healthy tissue. In this study, entropy, described as the randomness of pixel intensity, was considered a marker of tissue heterogeneity. Therefore, entropy was the most useful parameter in the differentiation between pathological and healthy tissues and it especially helped to differentiate even the earliest forms of OPSCC. The paper highlights how these analyses could have an important role in the evaluation of lesions not yet clearly visualizable by conventional radiological imaging.

Radiomics has also been used to discriminate OPSCC from other tumours that may affect oropharynx and tonsils, such as lymphoma. Compared to OPSCC, lymphoma shows very low apparent diffusion coefficient (ADC) values (0.4 − 0.7 × 10^−3^ mm^2^/s) on diffusion-weighted MR imaging because of the high cellular density due to the reduced extracellular space. Lymphoma also shows lower volume transfer coefficient (Ktrans) and extracellular volume ratio (Ve) values than OPSCC on dynamic contrast-enhanced perfusion MR imaging, due to the lower vascular permeability and extravascular-extracellular space. However, the diagnosis of lymphoma or OPSCC is often challenging to discriminate on morphological imaging alone [23,75,76,77,78].

Given these differences in their microscopic structure, in a study conducted by Bae et al. [53] it has been postulated that radiomics may show potential in discriminating between lymphoma and OPSCC. They identified on T1 post-contrast and T2 MRI sequences 19 features, 10 first-order features and 9 texture features capable of differentiating these two entities. More specifically, the diffuse heterogeneity of OPSCC is a characteristic that can be assessed with texture analysis. It is not found in lymphoma, given the general absence of areas of necrosis or colliquation in this latter type of tumour. The study by Mitamura et al. [63] evaluated low gray-level zone emphasis (LGZE) as the best feature (*p* = 0.004) on 18F-FDG PET/TC images to differentiate between SCC and non-Hodgkin’s lymphoma with a sensitivity of 55.6% and specificity of 88.0%, respectively.

For the staging of head and neck cancers, as well as those of the oropharyngeal area, the assessment of lymph node involvement is essential for correct loco-regional and distant staging. Although PET-CT is the first-line examination in the detection of pre-treatment lymphadenopathy, false positives due to inflammatory states and false negatives secondary to small lymph node sizes can occur [18,79,80].

Radiomics has also been used to provide supplementary information in addition to purely morphological imaging to distinguish benign lymph nodes from those with neoplastic infiltration in a study conducted by Park et al. [54] on the texture features of 204 lymph nodes. In this study, MRI, including ADC maps reconstructed by diffusion weighted sequences, showed that first and second level features (including those representatives of heterogeneity and shape) were statistically significant in discriminating between benign and metastatic nodes. This was attributed to the increased roundness of metastatic lymph nodes due the typical loss of their physiological ovoid shape. Six features (complexity, energy, global entropy, roundness, maximum probability, and short- run low gray-level emphasis) and five features (complexity, energy, global entropy, roundness, and maximum probability) were found to show statistically significant potential to differentiate all-sized and sub-centimetre-sized benign lymph nodes from metastatic ones.

CT imaging has also been used for this purpose with results almost comparable to MRI, underlining that texture analysis may be used to evaluate nodal involvement in patients with OPSCC tumours. In the study by Tomita et al. [62], 23 patients with a total of 201 cervical lymph nodes who underwent pre-operative CT with contrast medium followed by cervical neck dissection were evaluated. The lymph nodes, after histological evaluation, were randomly divided into a training cohort and a validation cohort. Three features, energy and entropy (GLCM), and zone length non uniformity (GLZLM) were significant in differentiating pathological lymph nodes. In addition to this, as previously mentioned, the first manifestation of a neoplasm of the oropharynx is often characterized by the presence of lymphadenopathy secondary to a primary occult tumour [81]. For this reason, in the diagnostic workup that follows an initial clinical evaluation, the aid of additional methods such as texture assessment may be helpful in the detection of occult tumours. A study by Lee et al. [55] focused on the application of radiomics on MRI imaging with the intent to study whether this new approach could help broaden the diagnostical possibilities of MRI. Texture analysis was used to help diagnose cases where clinical tonsillar swelling or radiological asymmetry were not sufficiently relevant on their own and, as a result of this study, shape features, 3D fractal analyses and moment features assessed on ADC maps were shown to have a diagnostic performance for palatine tonsil carcinomas that was comparable to F-FDG-PET/CT.

### 4.3. Use of Radiomics as a Prognostic Evaluation and in the Follow-Up of Oropharyngeal Cancer

Radiomics has also been used to assess which patients with locally advanced HPV+ OPSCC are at risk of developing distant metastases after radical chemo-radiotherapy. In a study conducted by Rich et al. [45], patients with primary tonsillar lesions were separated into two cohorts, restrospectively based on the subsequent development of distant metastases or not. A model was then produced using texture features related to tumour heterogeneity, especially a gray-level dependence matrix and a neighborhood gray-tone difference matrix with the intent of studying how well this model would be able to distinguish the two cohorts by radiomic signature only. Said model was found to be able to differentiate the two cohorts at a level of excellency and beyond with a median ROC area under the curve of 0.90.

Studying the phenotypic characteristics of a tumour can reflect the possible response to therapy, allowing for the evaluation of those neoplasms at a higher risk of non-response or recurrence. In the future, this could potentially enable personalised radiotherapy treatment by previously differentiating patients with the possibility of a good response to therapy or, on the contrary, at a higher risk of recurrence [82,83,84]. This task was carried out in a study performed in the M.D. Anderson Cancer Center, Houston, Texas, USA [85], in which 465 patients with histopathologically proven OPSCC patients were included. Texture analysis on the primary lesion was applied on pre-treatment enhanced CT and the population was divided following favourable or unfavourable clinical prognostic factors. The analysis of the extracted features—including GLCM, GLRLM, and neighborhood intensity different matrix—correlated with the distinction between the two risk groups of recurrence (high risk of recurrence and low risk of recurrence based on clinical factors). These patients then received follow-up at close intervals of about 2–3 months for the first two years and 3–6 months for the following years by imaging (contrast-enhanced CT, MRI or PET-CT scans) and eventual biopsy or reintervention in case of local failure.

The usefulness of radiomics features in regard to intra-tumoural and peritumoural assessment on CT imaging was also used on a cohort of 462 patients independently from HPV status to evaluate their prognosis in terms of disease-free survival (DFS) in a study by Bae et al. [53]. Three main features were selected (median, standard of sum-average, and median of mean intensity for HPV+ patients; median, skewness of sum-average, and kurtosis for HPV− patients) that correlated with a high statistical significance in discriminating between HPV+ and HPV− tumours. Consequently, through the implementation of a Cox regression model, a statistically relevant radiomic risk score for DFS in these two classes of patients was produced.

The prognostic role of radiomics applied to CT images was also evaluated in those patients who had undergone induction chemotherapy before radiotherapy in association with the assessment of size change of the primary tumour. Miller et al. [47] showed that the information gathered using texture analysis, specifically skewness and entropy, when factored in with the evaluation of percentage of tumour size change, resulted in a more precise assessment of patient prognosis than size evaluation alone with an estimated area under the curve (AUC) of 0.80 versus 0.56. A study conducted by Haider et al. [51] based on PET, CT or combined modality (PET/CT) imaging also evaluated how the use of quantitative imaging data, tissue density, texture features, lesion size and metabolic activity can be significant markers in identifying the tumour behaviour and creating prognostic and risk models for these patients. This study involved 311 patients. In particular, in HPV+ OPSCCs, the best radiomics-based model obtained a mean Harrell’s C-index ± standard deviation of 0.62 ± 0.05 (*p* = 0.02) for predicting progression-free survival. On the other hand, the value was 0.54 ± 0.06 (*p* = 0.32), using variables from the American Joint Committee on Cancer’s (AJCC) 8th edition staging scheme for survival prognostication and risk-stratification of HPV-associated OPSCC. This highlights how PET/CT analysis can provide complementary information to the AJCC scheme.

The study by Cheng et al. [65] focused on the assessment of tissue heterogeneity and therefore its prognostic impact in patients with T3/T4 OPSCC, identifying the feature zone-size non-uniformity (ZSNU) as an independent predictor of outcome and in particular, of PFS and DSS in patients with advanced stages of disease.

In another study, Cheng et al. [64] evaluated how lesion uniformity is strongly associated with survival outcome and differentiates responders from non-responders to therapy, and how a normalized gray-level cooccurrence matrix (GLCM) is an independent prognostic predictor in patients with OPSCC.

Some studies have also used MRI images to identify the prognostic value of radiomics features in assessing the outcome after therapy in patients with OPSCC. Some predictive models have been proposed based on the association of radiomics, clinical data, and primary tumour size both in patients with chemo-radiotherapy and surgical treatment [59,86].

By studying CT-based radiomics features, a study by Cozzi et al [60] assessed how some of them correlate with survival and local control after radiochemotherapy. The findings of this study reported that grey-level non-uniformity correlated significantly with the overall survival rate of the patients; that progression-free survival correlated with max value, compacity, and run-length non-uniformity; and that local control correlated with max value, volume, and small-zone high gray-level emphasis. According to their respective radiomic signatures, patients were subsequently divided into two categories of risk (high-risk and low-risk) to identify candidates for de-intensified therapy with reduced treatment-related toxicity and morbidity.

The difference in tissue homogeneity of the primary lesion, with hypervascular and other necrotic or hypovascular areas, may be reflected by the different texture features. This allowed attributed some of these features—including three histogram features and four gray-level run-length matrix features—to patients with an increased risk of local failure, as in the study conducted by Kuno et al. based on contrast-enhanced CT examinations either independently or combined with a [18F] FDG PET examination [48].

## 5. Conclusions

Texture analysis may lead to early differentiation between HPV+ and HPV− OPSCC and the detection of tonsillar lesions that may not be easily visualised by imaging techniques alone. Radiomics could also be an aid in the assessment of nodal metastases, tumour recurrences, persistence after combined radio-chemotherapy and prognosis. Based on our results, texture analysis is a useful additional tool for the detection of OPSCC in combination with currently used imaging techniques, such as CT, MRI and PET/CT. We believe that it will be increasingly used in the near future as an important support in diagnostic work-up and follow-up, potentially improving clinical decision-making.

## Figures and Tables

**Figure 1 cancers-14-02445-f001:**
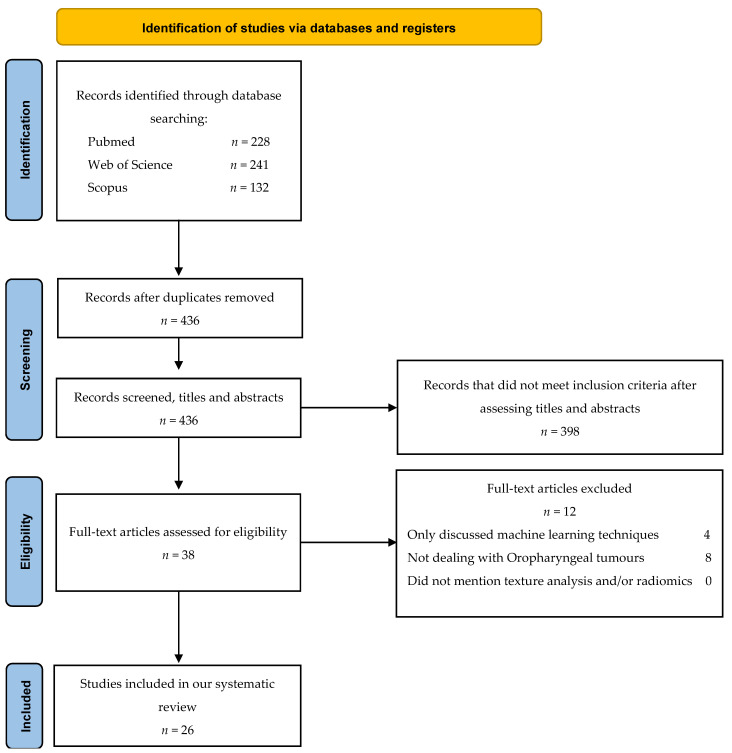
Flowchart consistent with preferred reporting items for systematic reviews (PRISMA 2020) statement.

**Table 1 cancers-14-02445-t001:** Search strategy.

Indexing Terms	Publications (N)
Pubmed	
#01 Head and neck carcinoma	60,209
#02 Head and neck carcinoma [MeSH terms]	44,169
#03 Oropharynx carcinoma	14,310
#04 Oropharynx carcinoma [MeSH terms]	9420
#05 Tonsil carcinoma	1716
#06 Tonsil carcinoma [MeSH terms]	1171
#07 Radiomics	5415
#08 Texture analysis	19,649
#09= #01 OR #02 OR #03 OR #04 OR #05 OR #06	68,885
#10= #07 OR #08	24,155
#11= #09 AND #10	228
Web of Science	
#01 Oral *	959,363
#02 Oropharyn *	30,346
#03= #01 OR #02	979,026
#04 Cancer *	3,463,149
#05 Carcinoma *	959,434
#06 Neoplasm *	211,729
#07= #04 OR #05 OR #06	3,849,511
#08 radiomic *	7657
#09 Texture analysis	79,538
#10= #08 OR #09	85,527
#11= #03 AND #07 AND #10	241
Scopus	
#01 Oropharynx	28,764
#02 Oral	1,263,096
#03 Oropharyngeal	25,707
#04= #01 OR #02 OR #03	1,292,826
#05 Cancer	3,486,977
#06 Carcinoma	1,251,963
#07 Neoplasm	1,073,234
#08= #05 OR #06 OR #07	4,127,810
#09 Radiomic *	6629
#10 “Texture Analysis”	14,579
#11= #09 OR #10	20,693
#12= #04 AND #08 AND #11	132

The research in Pubmed included both MeSH and free-text terms, whereas the searches in Scopus and Web of Science included only free-text terms or terms ending with an asterisk (*) which represents any number of letters (truncation). An additional manual search was performed using the reference lists of the examined studies. The searches were conducted on 16 February 2022.

**Table 2 cancers-14-02445-t002:** Studies included in the systematic review. SCC: squamous cell carcinoma, HNSCC: head and neck squamous cell carcinoma, OPSCC: oropharyngeal squamous cell carcinoma, NHL: non-Hodgkin’s Lymphoma.

Study	Sample Size	HNSCC Type	Histologic Type	Imaging Technique	Scans	Therapy
Kim T-Y et al., 2021 [49]	64	OPSCC	SCC of the Palatine tonsil	F-FDGPET/CECT	\	\
Buch K et al., 2015 [41]	40	OPSCC	HPV+ SCC	CECT	\	\
Fujita A et al., 2015 [42]	46	OPSCC	HPV+ SCC	CECT	\	\
Bogowicz M et al., 2017 [56]	93	HNSCC	HPV+ SCC	CECT	\	RTCT
Ranjbar S et al. [43]	107	OPSCC	HPV+ SCC	CECT	\	\
Leijenaar RTH et al., 2018 [57]	778	OPSCC	HPV+ SCC	CECT	\	RTCT
Yu K et al., 2017 [44]	315	OPSCC	HPV+ SCC	CECT	\	\
Choi Y et al., 2020 [52]	86	OPSCC	SCC	CECT	\	Untreated
Mungai F et al., 2019 [19]	50	OPSCC	HPV+ SCC	CECT	\	RT
Dang M et al., 2015 [61]	16	OPSCC	HPV+ SCC	MRI	Axial fast spin-echo T2-weighted imaging with fat saturation, axial fast spin-echo T1W1 with gadolinium, axial diffusion-weighted imaging	\
Bos P et al., 2021 [58]	153	OPSCC	HPV+ SCC	MRI	T1 weighted post contrast; post contrast 3dT1W	\
Bae S et al., 2020 [53]	87	OPSCC	HNSCC and lymphoma	MRI	Contrast-enhanced T1 and T2	\
Park J-H et al., 2019 [54]	36	HNSCC	Nodal metastases of SCC	MRI	ADC data of msEPI-DWI	\
Tomita H et al., 2021 [62]	23	OPSCC	Nodal metastases of SCC	CECT	\	\
Lee J-H et al., 2021 [55]	39	OPSCC	SCC of the Palatine tonsil	MRI	T1, T2, Contrast-enhanced T1, ADC	\
Rich B et al., 2021 [45]	225	OPSCC	Locally advanced HPV+ SCC	FBCT	\	Curative intentive RT or CT
Song B et al., 2021 [46]	582	OPSCC	HPV+ SCC	CECT	\	RT
Miller et al., 2019 [47]	38	OPSCC	HPV+ SCC	CT	\	Induction CT
Mes et al., 2020 [59]	323	HNSCC	SCC	MRI	T1 for feature extraction, STIR for segmentation	\
Kuno H et al., 2017 [48]	62	HNSCC	SCC	F-FDGPET/CECT	\	CT
Cozzi L et al., 2019 [60]	110	HNSCC	SCC	CECT	\	RT
Cheng N. M. et al., 2013 [64]	70	OPSCC	SCC HPV+	F-FDGPET/CECT	\	CTRT
Cheng N. M. et al., 2015 [65]	88	OPSCC	SCC	F-FDGPET/CECT	\	\
Haider S. P. et al., 2020 [50]	435 primary lesions 741 lymph nodes	OPSCC	SCC	F-FDGPET + non contrast CT	\	\
Haider S. P. et al., 2020 [51]	311	OPSCC	SCC	F-FDGPET/CECT	\	\
Mitamura K. et al., 2021 [63]	27 SCC 25 NHL	OPSCC NHL	SCC + NHL	F-FDGPET/CECT	\	\

**Table 3 cancers-14-02445-t003:** Significant features and extraction methods. DSS: disease-specific survival; GLCM: grey level co-occurence matrix; GLNU: Gray level non-uniformity; GLRLM: Gray-level run-length matrix; IQR: interquartile range; LRE: long run emphasis; LRLGE/LRHGE: long-run low/high gray level emphasis; LRLGLE: Long-run low gray-level emphasis; LZE: level zone emphasis feature value; LZHGE: large zone high gray level emphasis; LZLGE: large zone low gray level emphasis; MTV: metabolic tumour volume; NGLDM: neighbourhood gray level difference matrix; OS: overall survival; PFS: progression free survival; SD: standard deviation; SRHGLE: short run high gray level emphasis; SSF: spatial scaling factor; TLG: total lesion glycolysis.

Study	Segmentation	Relevant Texture Information	Software or Analysis Type	Free Software
Kim T-Y et al., 2021 [49]	ROI CT, VOI PET	Tumor side showed lower mean value for SSF 2-6, higher SD and entropy, lower skewness with SSF 0-4, higher kurtosis.	TexRAD and MIM software (software version unavailable)	No
Buch K et al., 2015 [41]	ROI CT	Histogram feature and histogram feature entropy show significant difference between HPV+ and − tumors.	In-house-developed using Mathlab (software version unavailable)	\
Fujita A et al., 2015 [42]	ROI CT	Mean, median, entropy, geometric mean, IQR; contrast, correlation, energy; LRHGE, skewness, kurtosis; L2, L5, L6, L7, L8 showed significant differences between HPV+ and – tumors.	In-house-developed (software version unavailable)	\
Bogowicz M et al., 2017 [56]	GTV defined for RT	A radiomic signature that correlate significantly with an increased local control was identified, while a more heterogeneous ct density distribution correlates with less local control.	In-house-developed (software version unavailable)	\
Ranjbar S et al., [43]	ROI CT	Histogram mean and entropy and GLCM entropy significantly differentiate between HPV+ and HPV− tumors.	OsiriX 6.5	No
Leijenaar RTH et al., 2018 [57]	GTV defined for RT	Radiomic analysis of images could help infer the molecular information of OPSCC	In-house-developed using Matlab 2014	\
Yu K et al., 2017 [44]	ROI CT	MeanBreadth and SphericalDisproportion correlate with HPV positivity in OPSCC	IBEX (software version unavailable)	Yes
Choi Y et al., 2020 [52]	Semiautomated ROI definition	Identification of a radiomic signature that correlates with HPV positivity; radiomics score and T staging associated with survival rate and prognosis	Syngo.via frontier software (software version unavailable)	No
Mungai F et al., 2019 [19]	VOI CT	Mean value; second order GLRLM (LRE, LRLGE, LRHGE, GLNUr, SRHGE); LZE, LZLGE, LZHGE, GLNUz, NGLDM PARAMETERS show variably significant correlation with HPV positivity in OPSCC.	LIFEx 3.40	Yes
Dang M et al., 2015 [61]	ROI MRI	Average value of local spectrum, SD of local spectrum, maximum value of local spectrum correlate significantly with p53 status of tumor.	OsiriX+ FTFT-2D tool (software version unavailable)	No
Bos P et al., 2021 [58]	ROI	Clinical evaluation and radiomic study correlate significantly with HPV positivity.	PyRadiomics 2.2.0	Yes
Bae S et al. 2020 [53]	Semiautomated ROI definition	There were 19 radiomics features selected as valuable for the distinction between HNSCC and lymphoma.	R software 3.5.1	Yes
Park J-H et al., 2019 [54]	ROI MRI	Complexity, energy and roundness features help discern reactive nodes from metastases in HNSCC.	IBEX (software version unavailable)	Yes
Tomita H et al., 2021 [62]	ROI CT	GLCM entropy, GLCM energy and diameter help discern reactive nodes from metastases in HNSCC.	LIFEx (software version unavailable)	Yes
Lee J-H et al., 2021 [55]	semiautomated VOI definition	The representative values of shape features, fractal analyses and moment features on ADC scans allow for a diagnostic performance of OPSCC of the palatine tonsil that is comparable to that of F-FDG-PET/CT	PyRadiomics 1.0	Yes
Rich B et al., 2021 [45]	GTV defined for RT	The model identified at a level of excellence the patients who went on to develop distant metastases.	SMOTE, ADASYN, borderline SMOTE (software version unavailable)	Yes
Song B et al., 2021 [46]	Manual ROI and GTV	There were 15 features that predicted HPV correlation; A 3 feature signature predicted DFS.	\	\
Miller et al., 2019 [47]	Manual ROI	Skewness and entropy features increase accuracy in progression prediction in patients treated with CT.	In-house-Developed (software version unavailable)	\
Mes et al., 2020 [59]	Semiautomated ROI definition	The integration of radiomic and clinical models outperforms the standard clinical prognostic model for HNSCC.	Velocity AI and In-house-developed software (software version unavailable)	\
Kuno H et al., 2017 [48]	Semiautomated ROI	Significant predictors of outcome of chemotherapy in patients with SCC were 3 histogram features and 4 gray-level run-length features.	In-house-developed MATLAB based software (software version unavailable)	\
Cozzi L et al., 2019 [60]	GTV defined for RT	A signature with 3 features was identified as predictive of overall survival in HNSCC; a 2 feature signature was predictive for local control.	LIFEx (software version unavailable)	Yes
Cheng N. M. et al., 2013 [64]	Semiautomated VOI selection	Age, tumor TLG, and uniformity independently associated with PFS and DSS; TLG, uniformity, and HPV positivity significantly associated with OS. New prognostic scoring system based on TLG and uniformity.	PMOD 3.3	No
Cheng N. M. et al., 2015 [65]	Semiautomated VOI selection	ZSNU identified as an independent predictor of PFS and DSS. Prognostic stratification system based on TLG, uniformity and ZNSU	In-House-Developed matlab based software (software version unavailable)	\
Haider S. P. et al., 2020 [50]	Manual ROI selection	PET-based radiomics signatures yield similar classification performance to CT-based models with a trend suggesting improved predictive performance when combined.	3D-Slicer version 4.10.1	Yes
Haider S. P. et al., 2020 [51]	Manual ROI selection	1037 PET and 1037 CT radiomic features quantifying lesion shape, imaging intensity, and texture patterns from primary tumors and metastatic cervical lymph nodes integrated to devise novel machine-learning models for OPSCC PFS and OS.	3D-Slicer version 4.10.1	Yes
Mitamura K. et al., 2021 [63]	Semiautomated VOI selection	SUVmax, MTV, and TLG did not differ significantly between the SCC and NHL groups. LGZE and HGZE significantly different between the SCC and NHL; LGZE the most discriminative (55.6% sensitivity, 88.0% specificity)	LIFEx (Software version unavailable)	Yes

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
