# Peer review of "Role of Texture Analysis in Oropharyngeal Carcinoma: A Systematic Review of the Literature"

_cancers, 2022, doi:10.3390/cancers14102445_

Round 1

Reviewer 1 Report

This is a nicely written review about the role of texture analysis for the diagnosis, prognosis and assessment of disease recurrence or persistence after therapy, early differentiation of HPV-positive versus HPV-negative cancer and the detection of cancer not visualised by imaging alone. 

The findings in CT and MRT are profoundly discrebed. The only censure I have is the absence of discussion of PET-associated features. I would like the authors to either add a PET-specific passage or explain why it is lacking. 

Reviewer 2 Report

  1. Please, in abstract delete this sentence "A systematic review was performed on 16 February 2022" and add last date search in the methods section
  2. Line 55-56 references are needed (for example, differences in HPVpos/neg tumours emerged in the following studies: https://doi.org/10.1038/s41416-020-0984-6 , 10.1038/nature14129 , 10.1158/1078-0432.CCR-09-0784 )
  3. Please include a more precise PICO question 
  4. Please report k-agreement among reviewers in the last phase of article inclusion
  5. I would suggest to replace the search table with the search string used in each database. For example (OSCC OR head and neck cancer OR OPSCC) AND (prognosis OR texture OR radiomics)
  6. Risk of bias assessment is missing
  7. Results in table should be written down in results sections. Result paragraph is poor in content and should include more information, for example, including most interesting outcomes and methods  among all included studies
  8.  

Reviewer 3 Report

The authors have tried to systematically review the role of texture analysis in oropharyngeal cancer . The topic is quite interesting but the way  authors have presented and discussed this work is not scientifically sound . I do not understand that why on the basis of abstract and title papers were chosen. Finally authors have only take 21 papers to create an hypothesis , which is quite a small data to suggest any clinical solution. The authors need to do proper literature search  to  reach on any conclusion. 

The authors did not suggest any novel conclusion through this manuscript. Now the days researchers are collecting the images of cancerous tissues and trying to create few guidelines that how looking at images of cancerous tissues and specially tissues of Oral cancer predictions can be made in terms of diagnostics about the severity of disease, prognosis of disease and personalized treatment of each OSCC patient. Authors should include those studies. The images from different studies should be made part of manuscript  and different images can be discussed followed by the correlation of those images with actual disease should also be presented. This will create a sound knowledge about this new field and will open a new avenue for researchers as well as clinicians. 

I do not understand the need for giving Table 1 & 2 separately , both can be merged and made in readable format rather than making table 2 unreadable. 

The discussion is too small and no new conclusions are proposed in this manuscript. 

Reviewer 4 Report

Dear Authors

This paper is well written but few editing requires. Please follow the suggestions for the betterment of this paper. 

1) In intorduction line 49-50 need reference.

2) line 92-93: authors can cite this litearture "https://doi.org/10.1016/j.jtumed.2021.05.012".

3) In method authors not discuss the PROSPERO. 

4) Discussion heading need major revision. Authors not wrote well. Even they can use professional English language editor support. 

5) Coonclusion is not well written. Need improvement. 

Round 2

Reviewer 2 Report

Point 2: Line 55-56 references are needed (for example, differences in HPVpos/neg tumours emerged in the following studies: https://doi.org/10.1038/s41416-020-0984-6 , 10.1038/nature14129 , 10.1158/1078-0432.CCR-09-0784)  

Response 2: We thank the reviewer for the suggestion. The literature has been cited accordingly: Augustin JG, Lepine C, Morini A, Brunet A, Veyer D, Brochard C, Mirghani H, Péré H, Badoual C. HPV Detection in Head and Neck Squamous Cell Carcinomas: What Is the Issue? Front Oncol. 2020 Sep 15;10:1751. doi: 10.3389/fonc.2020.01751. PMID: 33042820; PMCID: PMC7523032.

Response: Your statement canno't be covered by only one reference dealing with methodological issues in HPV infection ascertainment. I suggest including more references as indicated in the first round comment.

Point 4: "Kappa values of 0.01–0.20, 0.21–0.40, 0.41– 169
0.60, 0.61–0.80, 0.81–0.99, and 1 represented slight, fair, moderate, substantial, almost per- 170 "fect, and perfect agreement, respectively."

Please remove this sentence. 

Point 6: Results from risk of bias should be also written down in a paragraph and not only in table.

Point 7: Results section is still poor. Information from table 3 should be pooled together and written down in a paragraph, combining most commonly reported items/features and outcomes. 

Reviewer 4 Report

Dear Authors

revisions look good. Well done